# Cell death induced in glioblastoma cells by Plasma-Activated-Liquids (PAL) is primarily mediated by membrane lipid peroxidation and not ROS influx

**Sebnem Gunes**[1,2][☉]*, **Zhonglei He**[1,3][☉], **Evanthia Tsoukou**[1,2], **Sing Wei Ng**[2,4], **Daniela Boehm**[1,2], **Beatriz Pinheiro Lopes**[1,2], **Paula Bourke**[2,4], **Renee Malone**[1], **Patrick J. Cullen**[1,5], **Wenxin Wang**[3], **James Curtin**[1,2,6]*

**1** BioPlasma Research Group, School of Food Science and Environmental Health, Technological University Dublin, Dublin, Ireland, **2** Environmental Sustainability & Health Institute (ESHI), Technological University Dublin, Dublin, Ireland, **3** Charles Institute of Dermatology, School of Medicine, University College Dublin, Dublin, Ireland, **4** Plasma Research Group, School of Biosystems and Food Engineering, University College Dublin, Belfield, D4, Dublin, Ireland, **5** School of Chemical and Biomolecular Engineering, University of Sydney, Sydney, Australia, **6** Faculty of Engineering and Built Environment, Technological University Dublin, Dublin, Ireland

☉ These authors contributed equally to this work.
* james.curtin@tudublin.ie (JC); sebnem.gunes@tudublin.ie (SG)

## Abstract

Since first identified in 1879, plasma, the fourth state of matter, has been developed and utilised in many fields. Nonthermal atmospheric plasma, also known as cold plasma, can be applied to liquids, where plasma reactive species such as reactive Oxygen and Nitrogen species and their effects can be retained and mediated through plasma-activated liquids (PAL). In the medical field, PAL is considered promising for wound treatment, sterilisation and cancer therapy due to its rich and relatively long-lived reactive species components. This study sought to identify any potential antagonistic effect between antioxidative intracellularly accumulated platinum nanoparticles (PtNPs) and PAL. We found that PAL can significantly reduce the viability of glioblastoma U-251MG cells. This did not involve measurable ROS influx but instead lead to lipid damage on the plasma membrane of cells exposed to PAL. Although the intracellular antioxidative PtNPs showed no protective effect against PAL, this study contributes to further understanding of principle cell killing routes of PAL and discovery of potential PAL-related therapy and methods to inhibit side effects.

## Introduction

Cancer is a deadly disease that is the second biggest cause of mortality in Western nations [1]. Despite continued advancements in cancer therapy, new research areas are needed to improve clinical effectiveness. Combination treatments that attack tumour cells through multiple mechanisms, either concurrently or sequentially, rather than attempting to specifically target a

**Data Availability Statement:** The data used to generate plots and perform statistical analyses have been uploaded to the Open Science

Framework archive: https://doi.org/10.17605/OSF.IO/AGP49.

**Funding:** This work is supported by TU DUBLIN Fiosraigh Research Scholarship programme (S.G.), Irish Research Council Government of Ireland Postdoctoral Fellowship Award GOIPD/2020/788 (Z.H.), and EOSPG/2020/277 (B.L.), Science Foundation Ireland Grant Numbers 14/IA/2626 (P. B, P.C., J.C.), 15/SIRG/3466 (D.B.), 16/BBSRC/3391 (P.B.) and 20/US/3678 (P.B.) and 21/FFP-A/9189 (J.C.). The funders had no role in study design, data collection and analysis, decision to publish, or preparation of the manuscript.

**Competing interests:** The authors have declared that no competing interests exist.

single deregulated pathway have become popular in therapeutic development for oncology. Reactive oxygen and nitrogen species (ROS/RNS), and their associated redox signalling pathways affect a multitude of intracellular signalling pathways and are subjects of intriguing research [2].

Direct cold plasma produces long- and short-lived reactive species in the gas phase, such as nitric oxide, ozone, hydroxyl radicals, singlet oxygen, and superoxide anion [3]; however, plasma-activated liquid (PAL), as an indirect plasma system, is primarily composed of long-lived species such as hydrogen peroxide, nitrite, and nitrate. Although direct plasma treatment to cells and tissues is promising and approved in European dermatology centres [4], there is a growing demand to use plasma-derived ROS/RNS as putative injections in a more flexible manner which can be achieved by treating liquids or solutions with plasma devices [5]. The plasma-derived ROS/RNS are transferred from the plasma gas phase to the liquid phase in this process.

Li *et al.* (2017), investigated the use of dielectric barrier discharge (DBD) plasma in direct cancer cell treatment and a microwave-based plasma system in indirect cancer cell treatment. They made nitric oxide plasma-activated water (NO-PAW) as an indirect treatment. Results showed that NO-PAW created with a microwave plasma system had an 18% apoptotic effect on HeLa cells, while 4 minutes of DBD therapy had a 7% apoptotic effect [6]. The species found in PAW and plasma activated medium (PAM) have been reported to destroy cells via apoptosis while also boosting mammalian cell growth [7]. According to studies, the ROS species found in the PAW/PAM can cause the lipid bilayer membrane to peroxide, resulting in apoptosis [8].

Furthermore, according to Tanaka *et al.* (2011) PAM causes apoptosis in glioblastoma cancer cells via a caspase 3/7 route. Moreover, PAM treatment of cancer cells led to the downregulation of a survival signalling molecule (AKTkinase) [9]. When tumours are present in the body's internal organs, exposing malignant tissues to plasma torches/microjets for direct plasma treatment is difficult, if not impossible [10]. According to Saadati *et al.* (2018) indirect plasma treatment might be utilised extensively inside the body with far fewer adverse effects than direct treatment [11]. Subramanian *et al.* (2020) examined the effects of PAW on the viability of human breast cancer cells (MDA-MB-231) and healthy murine muscle-derived fibroblast cells using the MTT assay. According to the findings, the cell viability of MDA-MB-231 human breast cancer cells was reduced by over 75%. 6 and 12 minutes activation times of PAW diminished the cell viability of MDA-MB-231 cancer cells without affecting the vitality of healthy primary fibroblasts (murine muscle-derived fibroblast (MMF)), demonstrating selectivity [12].

Additionally, Boehm *et al.* (2017) generated PAW using a high-voltage DBD cold atmospheric plasma (CAP) technology and results showed high cytotoxic potential and superior storage stability. $H_2O_2$ concentrations can be linked to decreased cell viability and growth and used as an indicator of a plasma-activated water's potential efficacy. Results illustrated that PAW has a potency that exceeds $H_2O_2$-related toxicity compared to the standard $H_2O_2$ kill curves, indicating that other plasma-generated species are also important [13].

We wished to explore the specific mechanism of toxicity of PAL in Glioblastoma. Previously, we found that low dosages of PtNPs had non-to-low toxicity in the U-251 MG and HEK 293 cell lines while also having significant protective benefits against DBD CAP-induced cytotoxicity as an effective intracellular ROS scavenger [14]. We hypothesised that accumulation of PtNP in cells would allow us to differentiate between the role of extracellular and intracellular ROS in mediating cell damage and cytotoxicity. Therefore, in the current study, we investigate the roles of PtNPs when cancerous U-251 MG cells are exposed to indirect plasma treatment, using a previously reported spark discharge PAL generating device (Fig 1) [15].

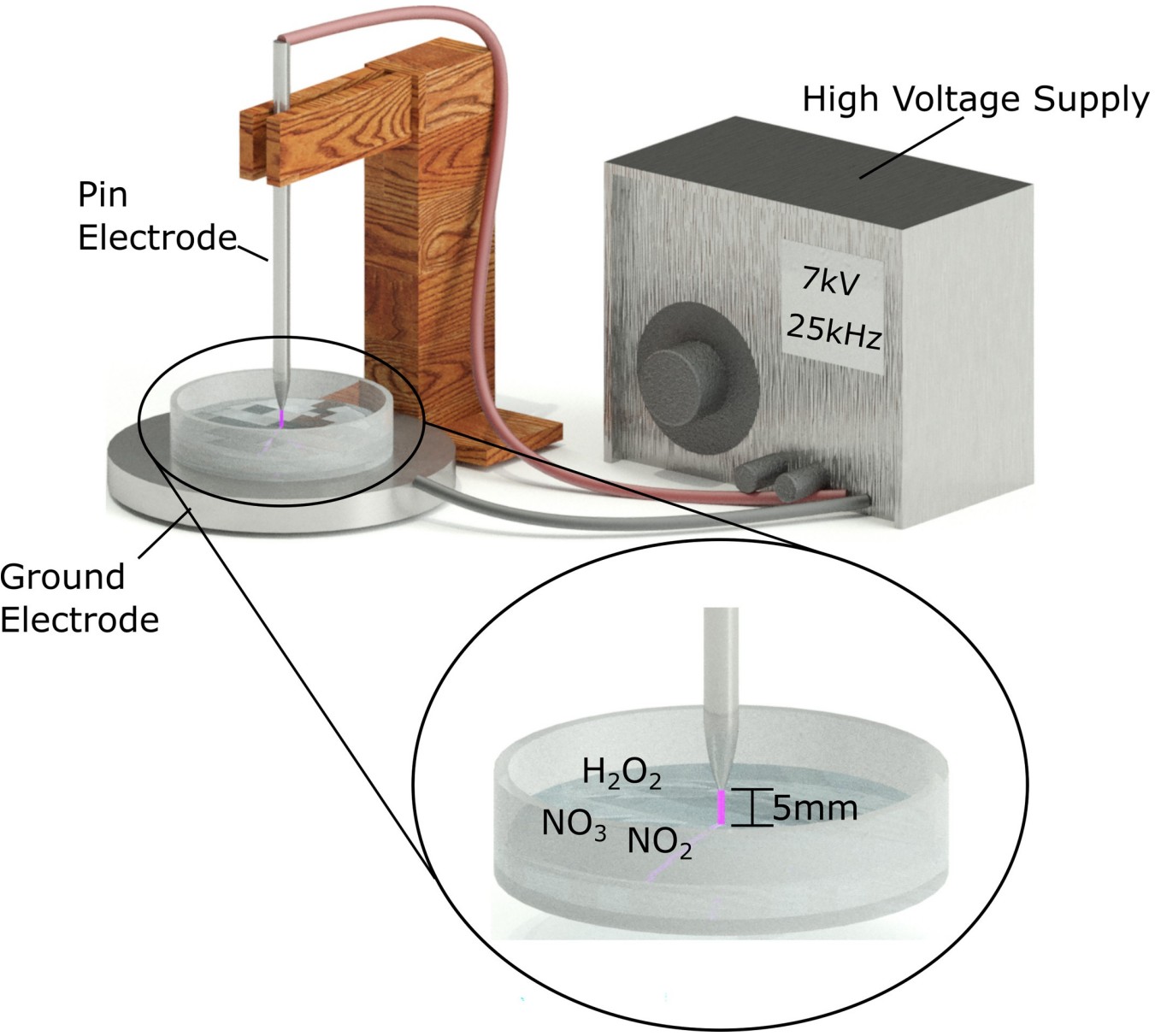

**Fig 1. A schematic illustration of the PAL generation device prototype used in this study.**

## Materials and methods

Human cancer cell lines (U251-MG) were used in the study. These are cell lines obtained from reputable commercial cell banks. Animal tissue (fetal calf serum) was used in the study. This was obtained from a reputable commercial company. The research project was approved by TU Dublin Research Ethics and Integrity Committee.

### Reactive Species Specificity (RSS) plasma system

An RSS Plasma system has been developed and previously characterised [16] and used in the "spark" configuration. Briefly, the high voltage electrode was a stainless-steel needle positioned perpendicular to the solution's surface, and the distance between the high voltage needle tip

and the liquid's surface was set to 5 mm. The plastic petri dish was put on a stainless-steel plate that was connected to the ground, and the discharge was operated in atmospheric air to get the spark setup. As a power supply, the PVM500 plasma driver (Information Unlimited Inc., USA) was employed, which has a maximum output of 20 kV and a variable frequency of 20–70 kHz. To generate plasma, a voltage of 7 kV was applied with a set frequency of 25 kHZ, and the system's applied power was 19 watt [16, 17].

## Plasma activated water treatment

A plastic petri dish (55 mm inner diameter) was filled with 10 mL of sterile deionised water to produce PAW, resulting in a water layer of around 4.2 mm depth, while the diameter of the electrode tip was around 1 mm (Fig 1). All PAW used in this study was created using a 10-minute generation time. The temperature of treated PAW has been determined to be around 60˚C using a thermal camera (S1 Fig). The PAW samples were cooled to room temperature, analysed, and stored at 4˚C in 15 mL tubes for 24-hour analysis. For cell treatment, PAW was used freshly or stored as indicated, and then diluted in fresh culture medium to specific % (v/v) concentrations, and then added to cells with various treatment conditions as indicated in each result section.

## Measurement of reactive oxygen species concentrations

Hydrogen peroxide concentrations in plasma activated water (PAW) were quantified using titanium oxy-sulphate (TiOSO4, Sigma-Aldrich, Arklow, Ireland) and spectrophotometric measurement. The calibration curve of known hydrogen peroxide concentrations (0, 82, 163, 326, 653, 979, and 1632 μM) was prepared by diluting 30% hydrogen peroxide standard solution (Perhydrol® for analysis EMSURE® ISO) and used to convert absorbance into hydrogen peroxide concentrations. Briefly, 10 μl of TiOSO4 solution and 100 μl of PAW or calibration curve samples were added to a 96-well microtiter plate. After 10 min incubation at room temperature, absorbance was read on a spectrophotometric plate reader at 405 nm [15].

Total oxidising species concentrations in PAW were quantified using oxidation of potassium iodide to iodine and spectrophotometric measurement as per method described in Boehm *et al.* (2016) with slight modification. The calibration curve of known hydrogen peroxide concentrations was prepared as described above and used to convert absorbance into total oxidising species concentrations. 50 μl of PAW or calibration curve samples, 50 μl of water and 100 μl of potassium iodide (1 M) were added to a 96-well microtiter plate. After 30 min incubation at room temperature, absorbance was read on a spectrophotometric plate reader at 390 nm [18].

## Measurement of reactive nitrogen species concentrations

Nitrite ion ($NO_2^-$) concentrations were measured by Griess reagent (Sigma-Aldrich, Arklow, Ireland). A range of known concentrations of sodium nitrite solution (0, 20, 40, 60, 80, and 100 μM) was used to prepare a $NO_2^-$ calibration curve and to convert absorbance into $NO_2^-$ concentrations. 50 μl of Griess reagent and 50 μl of PAW or calibration curve samples were added to a 96-well microtiter plate. After 30 min incubation in the dark, absorbance was read at 548 nm [19].

Nitrate ion ($NO_3^-$) concentrations were determined by 2,6-dimethyl phenol (DMP) using the Spectroquant® nitrate assay kit (Merck Chemicals, Darmstadt, Germany) using the manufacturer's instructions with minor modification. Samples were pre-treated with sulfamic acid to eliminate nitrite interference. A set of standard concentrations of sodium nitrate solution (0, 0.1, 0.25, 0.5, 1, 2.5, and 5 mM) was used to prepare $NO_3^-$ calibration curve and to convert

absorbance into $NO_3^-$ concentrations. Briefly, 200 μl of reagent A, 25 μl of PAW or calibration curve samples and 25 μl reagent B was added in order into a 1.5 ml microtube and then mixed vigorously. After 20 min incubation at room temperature, 100 μl of the total mixture was added to a 96-well microtiter plate and the absorbance was read at 340 nm [20].

## Cell culture

The human brain glioblastoma cancer cell line (U-251 MG) was provided by Dr Michael Carty (Trinity College Dublin). Cells were grown in Dulbecco's Modified Eagle's Medium-high Glucose (Merck) supplemented with 10% fetal bovine serum (Merck) and 1% penicillin (Thermo Fisher Scientific) and kept at 37˚C in a humidified incubator containing 5% (v/v) $CO_2$ (Sarstedt). The culture medium was changed every 2–3 days and cells were subcultured after reaching around 80% confluence. Using a 0.25% Trypsin-EDTA solution, cells were routinely subcultured in new flasks (Merck). Platinum, nanoparticle dispersion, 3 nm, was purchased from Merck.

## PtNPs and PAW treatment with Alamar blue assay

The Alamar blue assay was used to determine cell viability (Thermo Fisher Scientific). In 96-well plates (Sarstedt), U-251 MG cells were plated at a density of $1x10^4$ cells/well (100 μl culture media per well) and incubated overnight at 37˚C. The medium was removed, and a fresh culture medium containing 0–100 μg/ml PtNPs (Merck) was added and incubated for 1–24 hours as indicated in each section. Afterwards, the medium was removed again, and a fresh culture medium containing increasing concentrations ($0 \leq 30\%$) of PAW was replaced. Following a 2 to 48 h incubation with media containing PAW at 37˚C, the cells were rinsed once with phosphate buffered saline (Merck) and replaced with fresh media to continue incubation or do an Alamar Blue cell viability assay (10% Alamar blue in fresh media, incubate at 37˚C for 3 hours till reading fluorescence) at the end of the period as indicated in each section. Additionally, cells pre-incubated with PtNPs (0.032 and 4 μg/ml) for 24 h were treated with ROS generator 2,2-azobis (2-amidinopropane) dihydrochloride (AAPH) and $H_2O_2$ for 4 hours. For reading Alamar blue assay, fluorescence was measured (excitation, 530 nm; emission, 595 nm) by a Victor 3 V 1420 microplate reader (Perkin Elmer).

## $H_2DCFDA$ assay and flow cytometry

The generation of reactive oxygen species induced by CAP was detected using a cell permeable oxidant sensitive fluorescent dye 2,7-dichlorodihydrofluorescein diacetate ($H_2DCFDA$) (Thermo Fisher Scientific). U-251 MG cells were seeded in 35 x 10 mm Petri dishes (Sarstedt) at a density of $1×10^5$ cells/ml. After 24h, the growth medium was removed, and PtNPs (0.032 μg/ml) in medium were added and then incubated overnight at 37˚C. Medium was replaced with fresh medium containing 10% PAW and then incubated for 4 hours. Culture medium was replaced with fresh serum-free medium containing 25 μM H2DCFDA and cells were incubated for 30 min at 37˚C. Cells were washed with fresh medium once and then with PBS twice and all floating and attached cells were collected by trypsinisation. All liquids, including medium, washing PBS and trypsin-cell suspension, were transferred into one tube and centrifuged at 1200 rpm for 5 min. Cells were resuspended in PBS and the Beckman Coulter CytoFLEX Flow Cytometer was used to detect and measure fluorescence. Flow analysis was carried out with a 488 nm laser for excitation and FITC-A standard filter for $H_2DCFDA$ measurement.

## Lipid peroxidation and confocal imaging

The staining method of lipid peroxidation sensor C11-BODIPY was carried out as described by [21]. In brief, C11-BODIPY $^{(581/591)}$ (Thermo Fisher) stock solution was prepared in ethanol to 2 mM, then diluted in fresh culture media to 5 μM as the working concentration. Cells were seeded in 8-well Nunc™ Lab-Tek™ II Chamber Slide™ (Thermo Fisher) at 15,000 cells per well, settled overnight, and then incubated with fresh media with or without 0.032 μg/ml PtNPs for 24 hours. Subsequently, cells with PtNPs in the medium with/without Pyruvate were incubated with C11-BODIPY medium working solution at 37°C for 30 min and then washed twice with PBS before incubation with medium containing PAW for four hours.

Before observing under a Zeiss LSM 800 Airy confocal laser scanning microscope (University College Dublin, Conway Imaging Core), cells were washed twice with PBS and incubated with fresh medium containing working concentration (2 drops per ml) of NucBlue™ Live ReadyProbes™ Reagent (Hoechst 33342, Thermo Fisher) for 15 min at 37°C. Afterwards, cells were washed again twice with PBS and fixed by incubation with 4% paraformaldehyde in PBS (Merck) at 37°C for another 15 min. Finally, cells were washed once more with PBS prior to being mounted with 24 mm x 50 mm glass coverslips (Merck) and ProLong™ Glass Antifade Mountant (Thermo Fisher) and subsequent observation. The corresponding filter settings were as follows. NucBlue™, excitation 405 nm, emission 430–470 nm; C11-BODIPY (581/591), excitation 1: 488 nm, emission 1: 500–560 nm; excitation 2: 560 nm, emission 2: 560–620 nm.

## Statistical analysis

For each data point, three independent experiments were carried out in duplicate. Curve fitting and statistical analysis were performed using Prism 8 (GraphPad Software). Data are presented as a percentage, and all error bars are based on the standard error of the mean (S.E.M). Data points were verified using one-way ANOVA and Tukey's multiple comparison post-test to determine the significance, as indicated in figures (*P<0.05, **P<0.01, ***P<0.001, ****P<0.0001).

# Results

## The antioxidant effect of PtNPs against direct CAP exposure

The antagonistic effects between the direct DBD CAP treatment and PtNPs have been reported in a previous study and repeated here [14]. It demonstrated a 50% reduction in cell viability after 24 hours when U-251 MG cells were subjected to 50 seconds of CAP therapy, compared to controls. On the contrary, when cells were pre-incubated with PtNPs, they were protected from CAP-induced cytotoxicity, and lower concentrations of PtNPs alone did not cause any toxicity to cells in the absence of CAP treatment (Fig 2A). $H_2DCFDA$, a commonly used ROS indicator, was then used to study the intracellular ROS level after treatments. As seen in Fig 2B, ROS levels in cells increased considerably after direct CAP treatment compared with negative control. However, the mean fluorescence levels of oxidised $H_2DCFDA$ in CAP-treated cells were significantly decreased when they were pre-incubated with PtNPs, thus demonstrating the antioxidative effect of low doses of PtNPs. Based on these results, we selected two concentrations of PtNPs (0.032 and 4 μg/ml) further to study their potential interactions with indirect CAP treatment. In this study, spark discharge PAW generating device was used for indirect CAP treatment, which has been measured to generate, after running for 10 min, nitrate (~3.73 mM) and hydrogen peroxide (~0.88 mM) along with other yet explored oxidising species (~1.18 mM total oxidative species) with the set condition (S1 Table), whereas nitrite was not detectable. After 10 minutes of exposure to plasma for spark discharge (SD)

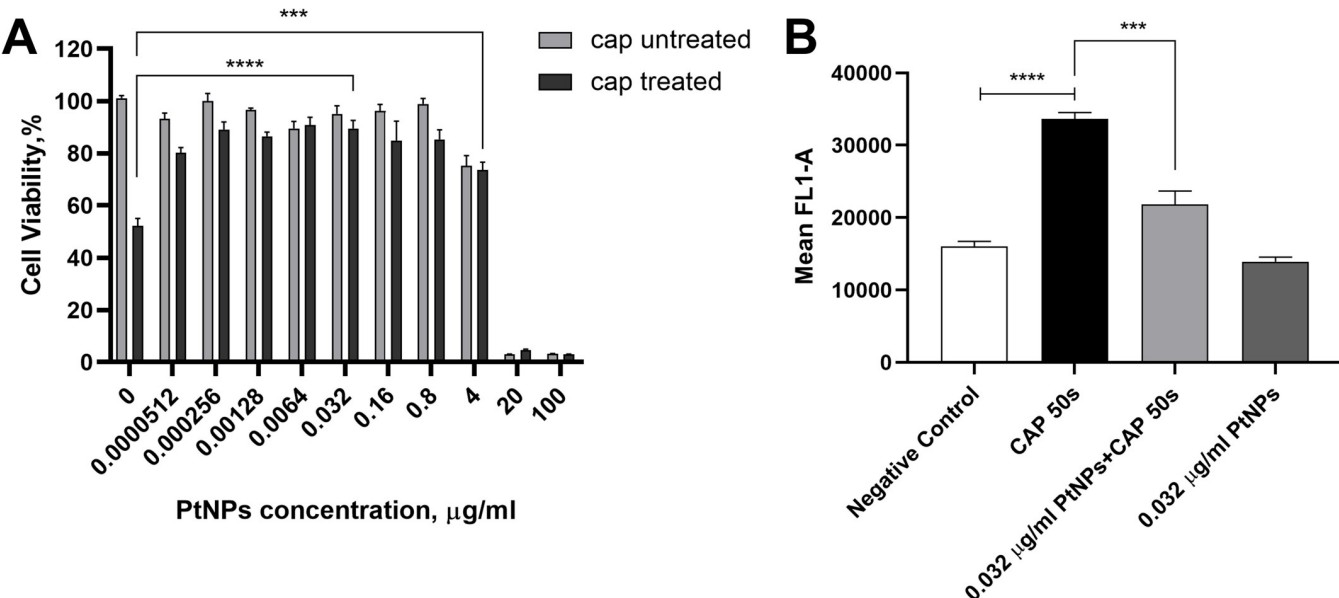

**Fig 2. Antioxidant effect of PtNPs against direct CAP treatment.** The experimental data was from our previous study [14]. (**A**) U-251 MG cells were incubated with increasing concentrations ($0 \leq 100$ μg/ml) of PtNPs for 24h before CAP treatment and Alamar blue analysis was carried out 24h after CAP treatment (**B**) Negative Control represents untreated cells, black CAP treatment and the others 0.032 μg/ml PtNPs with and without CAP treatment and data was acquired by flow cytometry.

PAW, the pH value changed from neutral to acidic, to a level of approximately 3, and remained steady after one week of storage. This spark discharge PAW also has been demonstrated with significant antimicrobial activity that is notably resistant to high temperatures [16].

## The role of Pyruvate on cell viability

Pyruvate is known as an antioxidant [18] and is common as an component in cell culture media. We sought to see if Pyruvate could affect PAW-induced cytotoxicity; therefore, we used two different culture mediums, with and without Pyruvate. As shown in Fig 3, dose-response curves were created utilising a variety of PAW percentages paired with two different non-toxic concentrations of PtNPs: 0.032 μg/ml and 4 μg /ml. Cytotoxicity was determined using an Alamar Blue cell viability assay. U-251 MG cells were pre-incubated with PtNPs for 24 h (Fig 3A and 3C) and 48 h (Fig 3B and 3D), and then the medium was replaced with a fresh medium containing freshly prepared PAW, and cells were incubated for 48 h. When cells grew in a medium with Pyruvate, there was no effect on the cell viability up to 20% PAW treatment in all conditions, and there was only a slight decrease at the highest PAW concentration (Fig 3C and 3D). On the contrary, as Fig 3A and 3B demonstrate, increasing percentages of PAW (20% and 30%) without pyruvate medium caused higher toxicity, killing almost all cells in the presence or absence PtNPs. Approximately 50% loss of cell viability was observed when U-251 MG cells were exposed to 10% and 15% PAW treatment (Fig 3A and 3B), although there is a slight increment in the cell viability with 48 h 0.032 μg/ml PtNPs incubation (Fig 3B). Cell viability was high with 5% and lower doses of PAW treatment in the presence or absence of PtNPs (Fig 3A and 3B).

## Simultaneous treatment of PAW and PtNPs to cells

To determine the simultaneous effect of PAW and PtNPs, we carried out two different methods. In the first method, U-251 MG cells were preloaded with the mixture of PtNPs (0.032 μg/

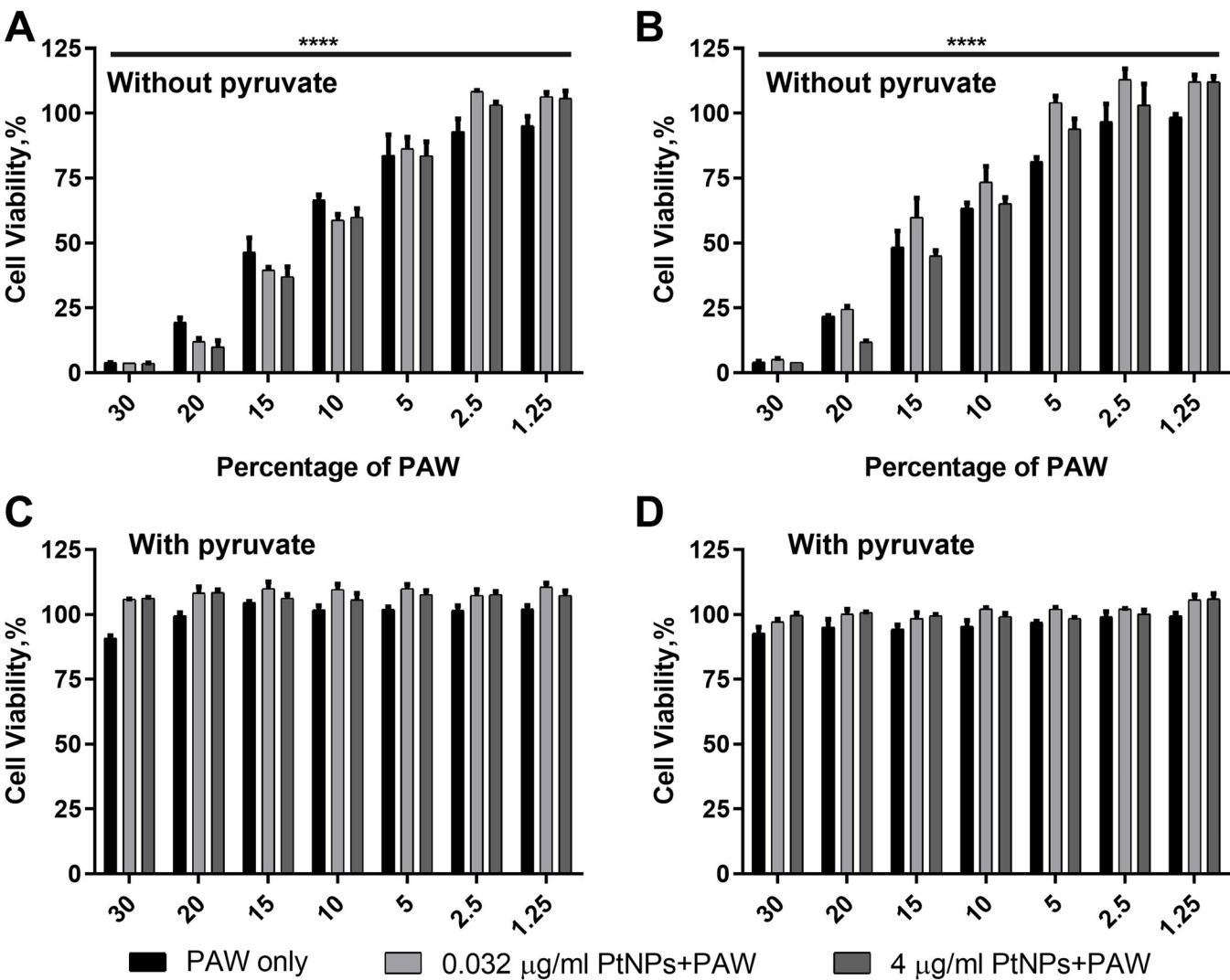

**Fig 3. Dose responses for PAW treatment.** In medium **without** Pyruvate, U-251 MG cells were incubated with 0.032 and 4 µg/ml PtNPs for 24 h (**A**) or 48h (**B**) and then treated with increasing concentrations (0 ≤ 30%) of PAW and incubated for 48 h; In medium **with** Pyruvate, U-251 MG cells were incubated with 0.032 and 4 µg/ml PtNPs for 24 h (**C**) or 48h (**D**) and then treated with increasing concentrations (0 ≤ 30%) of PAW and incubated for 48 h. Alamar blue cell viability assay was carried out 48 h after PAW treatment.

ml and 4 µg/ml) and a range of PAW concentrations (0 ≤ 30%) and then incubated for 24 h and 48 h at 37˚C. As a second method, we used higher percentages of PAW; 25, 30, 40, mixed each dose with 0.032 and 4 µg/ml PtNPs and incubated the mixture at 37˚C overnight and following that, cells were loaded with the mixture for 24 h and 48 h at 37˚C. As seen in Fig 4A and 4B, groups with higher doses of PAW (≥10%) with or without PtNPs had more significant toxicity. However, there is slightly higher cell viability with PtNPs mixture when compared to PAW only for groups with lower doses (≤5% PAW). Cell viability increased while the PAW doses decreased, and the viability reached 60% starting from 5% of PAW doses. A similar pattern was observed after treatment for 24 h and 48 h incubation in a medium without Pyruvate (Fig 4A and 4B). Interestingly, although cells were treated with higher percentages of PAW up to 40% and 0.032 µg/ml and 4 µg/ml PtNPs mixture, only 20% of toxicity for 24 h incubation and 10% for 48 h incubation was observed in medium without Pyruvate (Fig 4C and 4D).

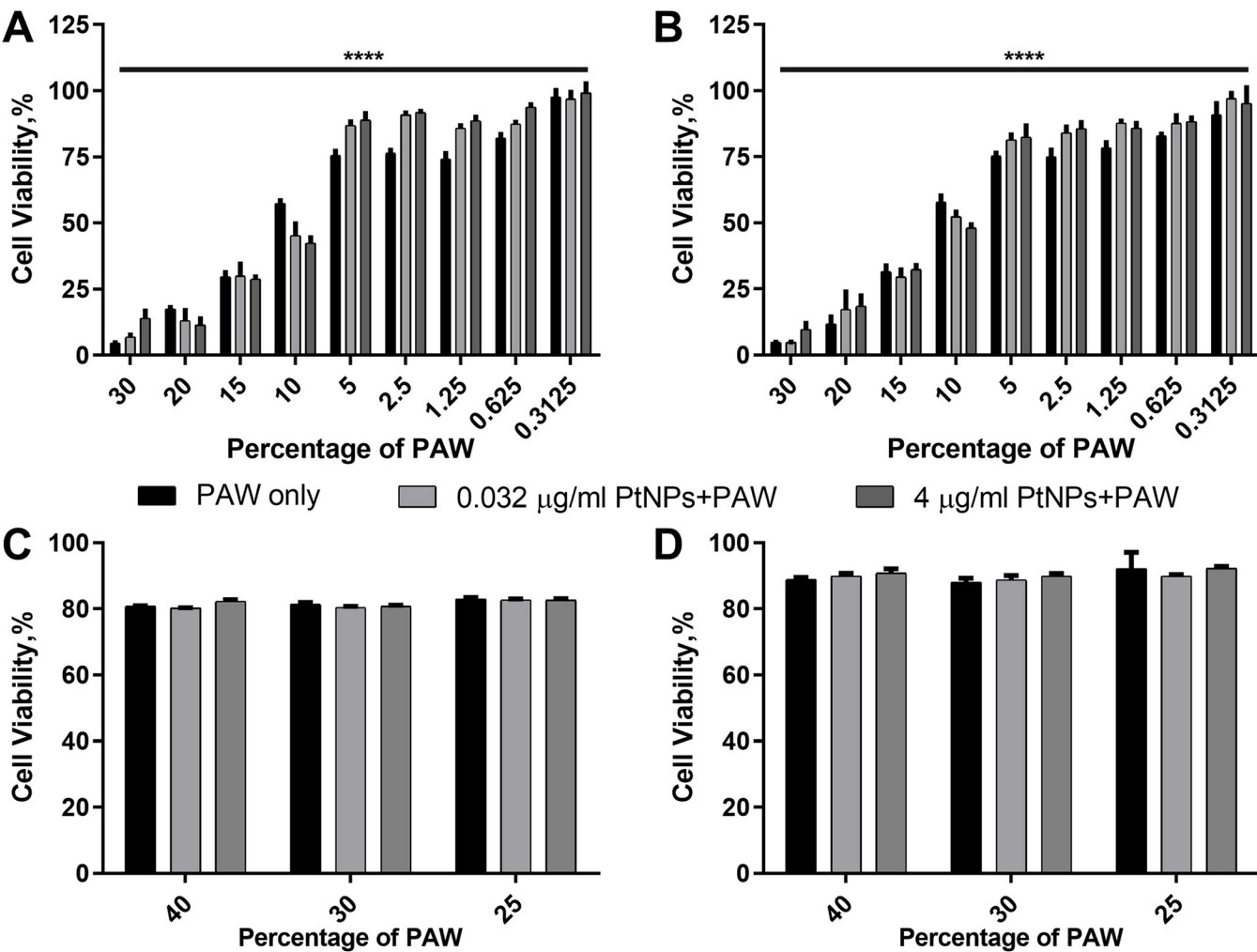

**Fig 4. Dose responses for simultaneous PAW treatment.** U-251 MG cells were treated with 0.032 and 4 µg/ml PtNPs and increasing concentrations
(0 ≤ 30%) of PAW simultaneously and incubated for 24h (**A**) or 48 h (**B**). 0.032 and 4 µg/ml PtNPs and 25, 30, 40% of PAW were mixed at incubated at 37˚C
for 24 h, U-251 MG cells were treated the mixture and incubated for 24 h (**C**) or 48 h (**D**).

## The difference between freshly prepared and 24 h stored PAW at 4˚C on cell viability

Interestingly, PAW incubated at 37˚C for 24 h showed minor cytotoxicity compared to freshly prepared ones with or without PtNPs. So, we wished to observe if there is any difference between freshly prepared and 24 h stored PAW at 4˚C on cell viability to elucidate reasons for the dramatically changed cytotoxicity of PAW following storage. We, therefore, treated U-251 MG cells with 0.032 µg/ml and 4 µg /ml PtNPs overnight and then replaced the medium without Pyruvate with fresh medium including 2.5, 5 and 10% PAW and incubated the cells for 48 h. Cytotoxicity was significantly higher when cells were treated with freshly prepared PAW at doses of 10% at around 80% (Fig 5A) compared to 24 h stored PAW, which caused approximately 50% loss of cell viability (Fig 5B). There was no significant difference when cells were treated with freshly prepared or stored PAW for the relatively lower doses, and cell viability was around 80% for both conditions.

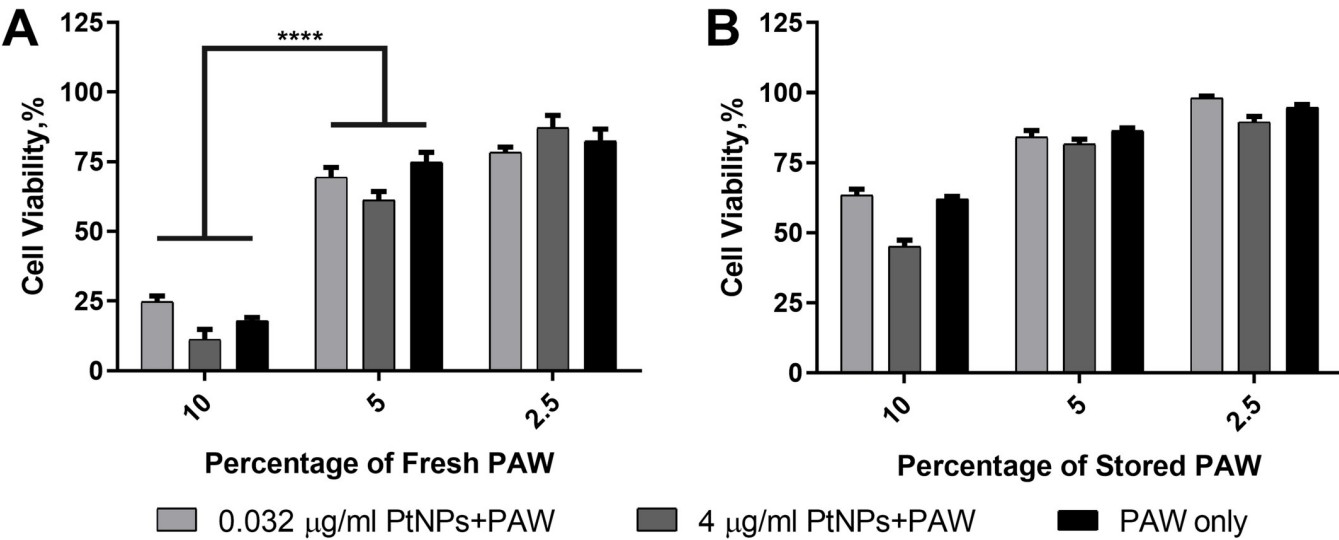

**Fig 5. Difference between freshly prepared or stored PAW treatment on cells.** U-251 MG cells were incubated with 0.032 and 4 µg/ml PtNPs for 24 h and then treated with 2.5, 5 and 10% freshly prepared PAW (**A**) or 24 h stored PAW (**B**). Alamar blue cell viability assay was carried out 48 h after PAW treatment.

### Time points assay for PAW

To examine the efficacy of PAW, we picked three different time points to add PAW to U-251 MG. Based on previous results and slight protection against PAW, we chose 0.032 µg/ml PtNPs to treat cells overnight. After 24 h incubation with PtNPs, cells were treated with PAW at 2.5, 5 and 10% doses for 2 h, 4 h and 24 h. As seen in Fig 6A, cell viability was 100% for the low doses of PAW; however, viability decreased almost 30% when cells were treated with only PAW at 10% dose, but 0.032 µg/ml PtNPs protected cells from 2 h of PAW treatment. Compared with 2 h PAW treatment, significant loss of cell viability was observed for 4 h (Fig 6B) and 24 h (Fig 6C) PAW treatment at a dose of 10% in the presence or absence of PtNPs. Although the cell viability was high at the lowest dose of PAW treatment, 24 h PAW treated cells showed less viability for the dose of 5% PAW compared to 4 h PAW treated ones (Fig 6B and 6C), which is in agreement with results in Fig 3.

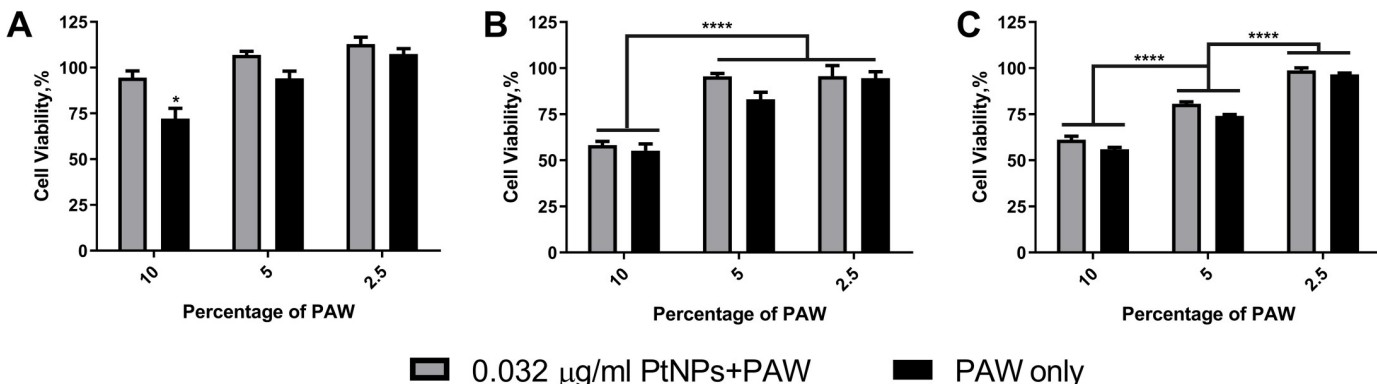

**Fig 6. Time points assay for PAW.** U-251 MG cells were incubated with 0.032 µg/ml PtNPs for 24 h and then treated 2.5, 5 and 10% PAW and incubated for 2 h (**A**), 4h (**B**) and 24h (**C**).

### Time points assay for PtNPs

Based on the results in Figs 3–6, we investigated the effects of 0.032 μg/ml and 4 μg /ml PtNPs at different time points. We treated U-251 MG with PtNPs for 1 h, 4 h, 8 h, 16 h and 24 h and then a range of PAW with increasing concentrations up to 30% was applied to cells and incubated for 48 h. Although cell viability was high for each PtNPs treatment time point for 5% and lower doses of PAW, for 30%, almost all cells were dead (Fig 7). As seen in Fig 7C, there is a slight decrease in cell viability at toxic doses of PAW (20% - 10%) compared to Fig 7A and 7B. When cells were treated with PtNPs for longer times, cell viability showed an increment with slight protection of PtNPs at high doses of PAW (Fig 7D and 7E). However, it is not possible to say that PtNPs protection is significant.

### Intracellular ROS level of PAW and PtNPs treated cells

In order to determine the effects of PtNPs on ROS generation by PAW (10%) U-251 MG cells were preloaded with $H_2DCFDA$. ROS levels in cells following PAW treatment remained at the almost same level as the negative control (Fig 8A). Additionally, PAW-treated cells pre-incubated with PtNPs (0.032 μg/ml) and cells pre-incubated with PtNPs only showed the same mean fluorescence levels (Fig 8). In contrast, AAPH-dependent ROS production and $H_2O_2$ brought a significant increase in intracellular ROS. The mean fluorescence levels of 25 mM ROS-generator and 2 mM of $H_2O_2$ were increased by a factor of 3 and 2 times, respectively, compared to the negative control (Fig 8).

### Cellular membrane lipid peroxidation of PAW and PtNPs treated cells

The potential damage pathway of PAW treatment insusceptible from PtNPs was then investigated via confocal microscope and membrane peroxidation sensor C11-BODIPY (581/591). C11-BODIPY (581/591) is a lipophilic dye that intercalates cellular lipid membranes and shifts their emission peak from around 590 nm (Red) to around 510 nm (Green) after being oxidised, thus is commonly used as a sensitive lipid peroxidation sensor.

As seen in Fig 9, with Pyruvate in the culture medium, no significant lipid peroxidation was observed in the PAW untreated group, but a slight signal in PAW treated group (Top two rows), which agrees with results demonstrated in Fig 9C and 9D. However, 4 hours of incubation with 10% PAW led to significant lipid peroxidation when the culture media without Pyruvate was used, despite potential protective antioxidant effect of PtNPs (Fig 9 bottom two rows), which also agrees with our results in Figs 3–7 that PtNPs surprisingly showed no significant protective effect against PAWs treatment.

## Discussion

The uses of cold plasma for medical purposes have been explored much more recently, especially for cancer therapy. Several methods of plasma treatments have been considered highly promising to eliminate cancer cells via generated reactive species and other effects, including plasma jet and plasma-activated liquids [22]. Plasma jet, as the representative technique for tissue treatment, has been demonstrated to suppress cancer cells effectively [23]. An in-depth understanding of the cancer-killing effects has been established, and several pro-oncogene or tumour suppressor-dependent regulations of antioxidant/or ROS signalling pathways were discovered to be related, as introduced in detail [23]. On the other hand, plasma jet technology also has been well developed, especially cold atmospheric pressure air plasma jet (CAAP-J), which is considered one of the most promising plasma treatment methods for medical uses, and an extensive review article is also included here for further information [24].

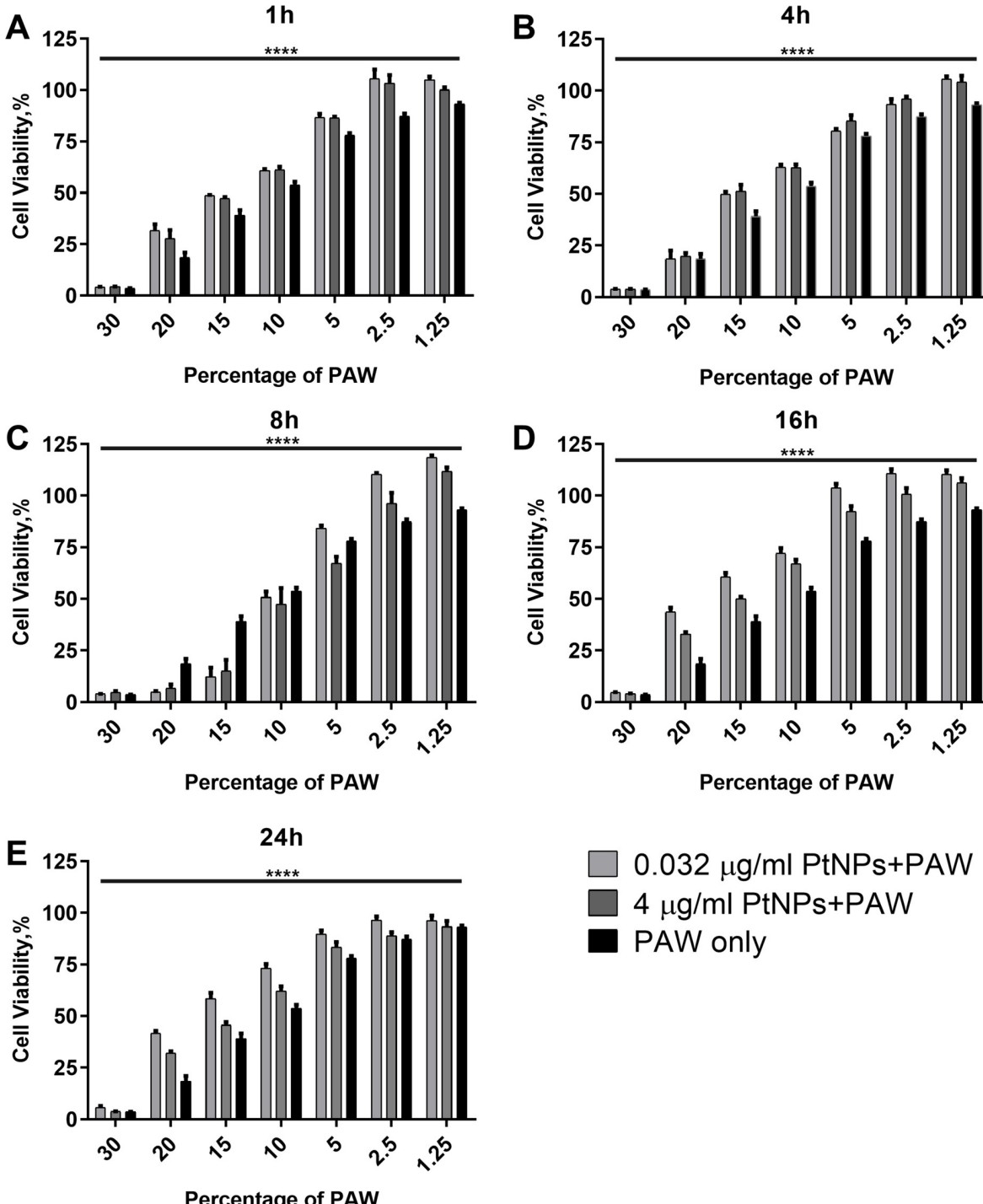

**Fig 7. Dose responses for PtNPs treatment.** U-251 MG cells were incubated with 0.032 and 4 μg/ml PtNPs for 1h (**A**), 4h (**B**), 8h (**C**), 16h (**D**), and 24h (**E**) then treated with increasing concentrations (0 ≤ 30%) of PAW. Alamar blue cell viability assay was carried out 48 h after PAW treatment.

Plasma-activated liquids (PALs) have been studied extensively as promising methods to manipulate ROS for therapeutic purposes due to their unique physicochemical properties. The dissolution of gas plasma species into liquid causes dynamic chemical reactions, resulting in a

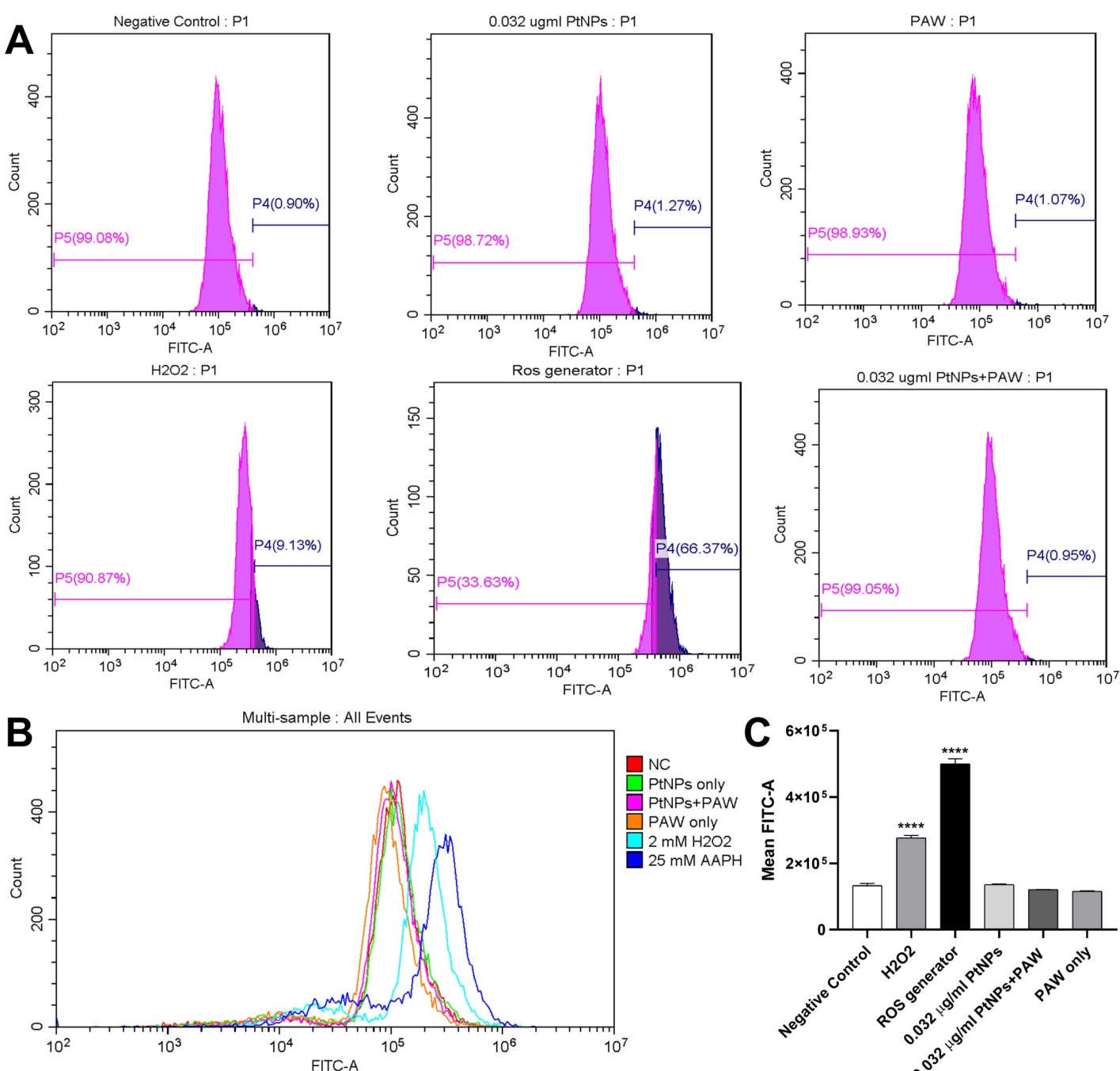

**Fig 8. Intracellular reactive oxygen species production in U-251 MG.** U-251 MG cells were incubated with 0.032 μg/ml PtNPs for 24 h and then treated with 10% PAW. (**A**) Histograms representing negative control, 0.032 μg/ml PtNPs, 0.032 μg/ml PtNPs combined with 10% PAW, 10% PAW only, H$_2$0$_2$ and ROS generator as positive controls. (**B**) Multi-comparison of ROS production in U-251 MG cells. (**C**) Mean fluorescence level of oxidised H2DCFDA in cells.

sequence of aqueous reactive species, some of which are short-lived transient reactive species that are difficult to identify [25]. While the total biological effects of PALs may be caused by a variety of short and long-lived species, it is well known in the field of plasma medicine that H$_2$O$_2$, NO$_2^-$ and NO$_3^-$ are detected as stable, long-lived species in PALs that play an essential role in biological effects [26]. Furthermore, PALs have been noted with particular toxicity

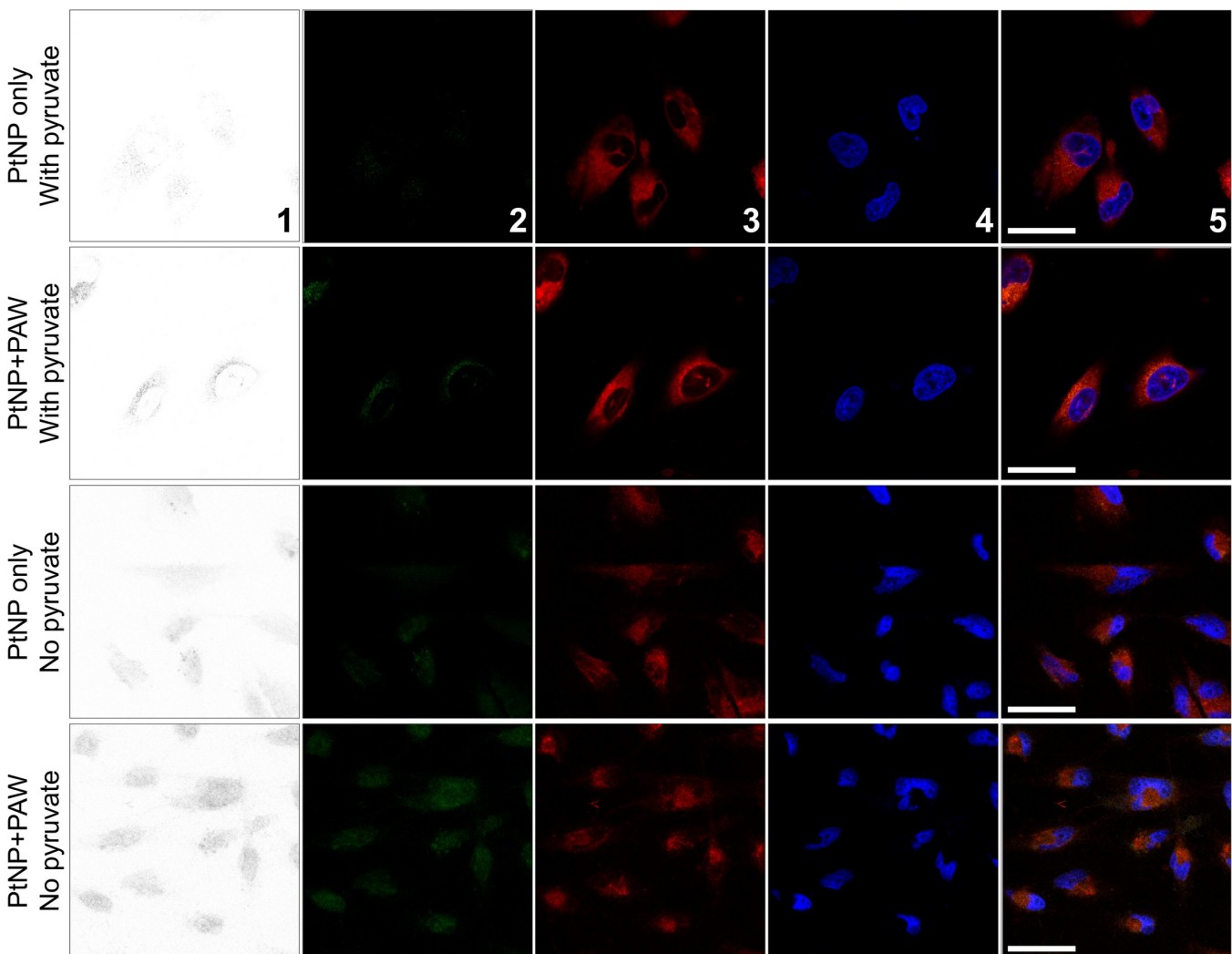

**Fig 9. Lipid peroxidation assay.** C11-BODIPY (581/591) staining shows increased lipid peroxidation induced by four hr treatment of 10% PAW in media without Pyruvate. Confocal imaging of C11-BODIPY loaded cells, 1st to 5th panels: Inverted, highlight oxidised form of C11-BODIPY using ImageJ; Green, oxidised form of C11-BODIPY; Red, reduced form of C11-BODIPY; Bule, NucBlue staining showing nucleus; Merged images (blue, green and red channels); Scale bar (from top to bottom, 30 µm, 30 µm, 50 µm, 50 µm).

leading to cancer cell death. For instance, Yan et al, demonstrated that pancreatic cancer cells and glioblastoma cells were specifically vulnerable to plasma-activated medium and plasma-activated Phosphate-buffered saline respectively [27].

PtNPs can be naturally taken up and accumulated into cells. Moglianetti *et al.* (2016) investigated PtNPs internalization and subcellular localisation. HeLa and Caco-2 cells were subjected to Inductively Coupled Plasma Atomic Emission Spectroscopy (ICP-AES) studies to determine the cellular uptake of PtNPs 24 hours after incubation, at 50 µg/mL concentrations. According to experimental findings, both 5 nm and 20 nm PtNPs accumulated internally in substantial amounts, with the smaller NPs showing the greatest uptake. Additionally, 5 nm PtNPs demonstrated four times higher uptake than 20 nm PtNPs in both cell lines in terms of Pt quantity [28]. Teow *et al.* (2010) treated HeLa, MCF7 and IMR90 cells with 100 µg/mL of 13.5 and 91.3 nm PtNPs and observed the scattered light from NPs showing bright spots under CytoViva optical microscopy [29]. Pelka *et al.* (2009) detected the uptake of 100 nm PtNPs or

smaller sizes at a concentration of 1000 ng/cm$^2$ in the HT29 cells incubated for 24 h by scanning electron microscopy (SEM) combined with the technique of focused ion beam (FIB) milling [30]. Elder *et al.* (2007) observed intracellular accumulation of 20 and 35 nm of PtNPs at concentrations of 25 and 250 μg/mL in HUVEC cells using TEM [31].

This study investigated the PtNPs (3 nm) as a potent intracellular scavenger of ROS, combined with PAW therapy on the U-251 MG cancer cell line. As previously established, low dosages of PtNPs showed non-to-low toxicity against U-251 MG cell lines with intracellular accumulation while demonstrating considerable protective benefits against cytotoxic effects induced by treatments of the direct DBD CAP system (Fig 2), which agree with other studies for antioxidative PtNPs [14, 32–36]. Therefore, we employed 0.032 μg/ml and 4 μg/ml of PtNPs based on previous data showing that concentrations of PtNPs at or below 4 μg/ml did not lead to significant cytotoxicity for the U-251 MG cell line.

We first identified the effects of the common antioxidant component, Pyruvate, in a cell culture medium. The cell viability rapidly deteriorated when the concentration of PAW was increased to 10% and higher doses when Pyruvate was omitted from the culture medium (Fig 3A and 3B). However, cell viability remained high even at high doses of PAW in the presence of Pyruvate (Fig 3C and 3D), which demonstrated that the cytotoxicity effect caused by fresh PAW could be suppressed by antioxidative Pyruvate and thus, the cytotoxicity of PAW can be confirmed to be highly related with ROS. Furthermore, surprisingly, no significant antioxidative effect of PtNPs was observed, except a slight increment in the cell viability with those groups pre-incubated with 0.032 μg/ml PtNPs for 48 h against PAW treatment (Fig 3B).

Furthermore, it has been demonstrated that other culture medium components may also scavenge ROS. For instance, it has been demonstrated that the generation of •NO only was observed in plasma-treated water but not in DMEM with or without FBS, which can be due to D-Glucose and other compounds still to be identified [37]. The presence of FBS, the composition of which may vary between batches and be hard to identify at the current stage, may also contribute to ROS scavenging [37]. Therefore, we have determined the levels of reactive species (nitrate and total oxidative species) in all representative cell treatment conditions (S3 Fig) to assist further discussion below. As seen in S3 Fig, we diluted PAW to various solutions to 30% before measurement. Interestingly, dilution in cell culture media with or without pyruvate did not change the nitrate level compared to 30% PAW in water and showed a linear decline compared to 100% PAW. PtNPs did not affect nitrate levels, which matches our previous results (S3 Fig). On the other hand, the total oxidative species level decreased significantly in culture medium without pyruvate, compared to 30% PAW in water. Furthermore, the pyruvate completely scavenged peroxides in the culture medium and the same trend was observed with the ROS generator too, which agreed to Fig 3. The presence of PtNPs did not affect total oxidative species level either (S3 Fig).

Then we wanted to seek if there is any synergistic or antagonistic effect of PtNPs and PAW when they are added to the cells simultaneously. As Fig 4A and 4B present, we detected significant cytotoxicity when high percentages of PAW were used, which agrees with the results in Fig 3. Meanwhile, no significant protective effect of PtNPs was observed for simultaneous treatment. Although we noted that PtNPs seem to slightly increase cell viability at most low PAW concentrations, it was not significant and became more inconsistent at higher PAW concentrations. The 10% PAW point in Fig 4A and 4B showed a contrary tendency that PtNPs slightly decreased cell viability, which was observed in Fig 3A, Fig 5A and 5B at higher PAW concentrations. However, the weak trend was not consistent in repeated experiments (S4 Fig). Interestingly, when PAW was incubated overnight at 37°C with or without PtNPs, it lost the cytotoxic effect even at the highest doses of 25, 30 and 40% (Fig 4C and 4D), which may demonstrate the instability of generated ROS or other cell-killing factors in PAW. To confirm the

reason for changes in cytotoxicity of PAW after storage, PAW was prepared freshly or stored at 4°C for further comparison (Fig 5). PtNPs were not found to increase cell viability; however, there is significant cytotoxicity when cells were treated with especially freshly prepared 10% PAW combined with/without PtNPs. Similarly, Shen *et al.* (2016) investigated how different storage temperatures (25°C, 4°C, -20°C, and -80°C) affected the bactericidal activity of PAW against *S. aureus* and the physicochemical parameters of PAW after 30 days. Their data revealed that the bacterial activity of PAW held at 25°C, 4°C, and -20°C reduced over time and was influenced by three germicidal factors: oxidation-reduction potential (ORP), $H_2O_2$, and $NO_3^-$. Furthermore, PAW held at -80°C retained bactericidal action, with $NO_2^-$ contributing to this ability in conjunction with $H_2O_2$ [38].

Lastly, to further confirm that PtNPs, although as a potential antioxidant, cannot suppress the cytotoxicity effects of PAW, we tested the cell viability at different times after PAW treatments (2 h, 4 h, 24 h) and three different doses of PAW (2.5, 5 and 10%) in the absence or presence of 0.032 μg/ml and 4 μg/ml PtNPs. Mohades *et al.* (2015) measured the vitality of the SCaBER cancer cell immediately after plasma-activated medium (PAM) administration (0 h), as well as 12 h, 24 h, and 48 h afterwards. According to the results, PAM did not influence cell survival immediately after application; on the other hand, the data after 12 hours, 24 hours, and 48 hours after PAM treatment revealed that when cells were exposed to PAM for longer durations, the cell viability was reduced up to 90% [39]. In agreement with this study, we found that 2 hours following PAW administration had no effect on cell viability with/without PtNPs (Fig 6A). However, viability was reduced after 4 h and 24 h of exposure to PAW in the presence or absence of PtNPs (Fig 6B and 6C). Still, we only observed slight protection effect of 0.032 μg/ml PtNPs pre-incubation against 2 h of 10% PAW treatment with around 20% increase of cell viability (Fig 6A), but not in other groups.

Our results demonstrated that PtNPs confer minimal to no protective effect against PAW, which may be related to the cell-killing mechanism of PAW treatment used in this study and was surprising compared with our previous results when investigating the protective effect of PtNPs against direct plasma treatment, and other studies regarding the antioxidative effect of PtNPs [14, 32–36]. Several *in vitro* and *in vivo* experiments revealed that pure PtNPs or PtNPs coated with cell compatible materials have low harmful effects [40]. PtNPs are usually thought to be less hazardous than silver NPs and to have outstanding antioxidant properties, lowering the generation of ROS [41]. Therefore, we hypothesise that it could potentially be that the ROS damage caused by the relative mild PAW treatment used in this study, unlike direct plasma treatment between two electrodes, is mainly limited to the cell surface and therefore not affected by intracellularly accumulated PtNPs, until there is a leakage of the cell membrane and influx of ROS. Meanwhile, the co-incubation of PtNPs and PAW in the medium also do not present reduced cytotoxicity, which agrees with our previous study that sufficient accumulation inside cells and high concentrations of PtNPs are critical for its antioxidative efficacy.

In several previous studies, hydrogen peroxide has been reported as the principal cytotoxic reactive species in PAL, while the reduction of cell growth and viability had linear correlations to the reduction of hydrogen peroxide concentrations in different PALs [16, 18, 20]. In contrast, the addition of nitrite and nitrate (up to 1.2 mM) showed no cytotoxic effects on U-251 MG cells in another study [13]. However, various damaging pathways of PAL are yet fully explored. To confirm this hypothesis of this particular PAW cytotoxic pathway insusceptible to antioxidative PtNPs, we investigated intracellular levels of ROS and lipid membrane peroxidation caused by PAW in the presence of PtNPs, using flow cytometry and confocal microscopy.

$H_2DCFDA$ has been used as an intracellular ROS indicator, and as predicted, no significant increased signal of intracellular ROS was observed when cells were treated with 10% PAW for

4 hours (Fig 8). However, as positive controls, high concentrations of hydrogen peroxide solution and ROS generator more aggressively induced significant increases of fluorescent signal showing intracellular oxidation (Fig 8), which likely was due to the influx of hydrogen peroxide via the damaged porous membrane. Despite no detectable increase of intracellular ROS level, incubation with 10% of PAW 2 mM hydrogen peroxide for 4 hours showed a decrease of 40–50% of cell viability, whereas 2-hour treatment of 10% PAW only caused minor cytotoxicity. Thus, the potential reason for reduced cell viability by 10% PAW 4 hours of treatment, which was relatively mild compared to positive control ROS solutions, was investigated using confocal imaging and lipid peroxidation sensor C11-BODIPY. As seen in Fig 9, the presence of pyruvate inhibited notable lipid peroxidation, which agrees with the results in Fig 3C and 3D. Pyruvate, as an effective antioxidant and common component in cell culture medium, can effectively eliminate cytotoxic effects of PAW, in which ROS play a central role. The potential antagonistic effects of pyruvate and other potential antioxidants, artificially or naturally existing in *in vitro* reagents and *in vivo* internal environment, thus should be noted in studies that focus on the roles of PAL and other ROS-generating plasma devices. The observation of significant lipid peroxidation in groups treated with PAW in the absence of pyruvate (Fig 9) then further proved our hypothesis that membrane lipid peroxidation induced by ROS in relative mild PAW treatment is sufficient to lead to significant reductions in cell viability and final cell death with no need of further membrane perforation or ROS influx. Over shorter time periods, it is therefore likely that the influx of ROS into cells exposed to PAW is buffered by an intracellular antioxidant mechanism, whereas accumulating damage to external membranes can continue to take place. Therefore, PtNPs, accumulated inside cells as an antioxidant, do not protect cells from the initial damage caused by PAW. Interestingly, after 4 h, oxidised lipid staining in cells displayed a punctate pattern, indicating that membrane internalisation, turnover and recycling had been activated as the cell attempted to repair the damaged plasma membrane.

In summary, whereas the potential inhibition and cytotoxic effects of nitrite, nitrate [42] other radicals and their byproducts, such as peroxynitrite/peroxynitrous acids, that can generate in PAL cannot be denied [16, 43], hydrogen peroxide and other ROS have been demonstrated to play the principal role in the cytotoxic effects caused by spark discharge PAW. Furthermore, the membrane lipid peroxidation caused by ROS in relatively mild PAW treatment is sufficient to cause a significant reduction in cell viability and final cell death, without further membrane damage and the influx of ROS via the porous membrane, thus cannot be prevented by PtNPs or other similar intracellular antioxidants. Lastly, we also tested the cytotoxicity of $H_2O_2$ and ROS generator in the presence of PtNPs. Miriam *et al.* (2020) established an assay using a variety of $H_2O_2$ concentrations to measure oxidative stress in the porcine small intestinal epithelial cell line IPEC-J2 using CM-$H_2$DCFDA assay. Results showed that 2.5 and 3 mM of $H_2O_2$ significantly increased intracellular ROS levels however 1 to 2 mM $H_2O_2$ did not cause an increase in intracellular ROS levels in the cell [44]. In another study, Engelmann *et al.* (2005) anayzed GSH concentrations and ROS formation using $H_2O_2$ as a positive control at a concentration of 0.02, 0.2 and 2 mM for 90 min in primary human fibroblasts (HPF). At concentrations of 0.2 mM or higher, the results showed that $H_2O_2$ enhanced intracellular ROS and decreased GSH. With higher $H_2O_2$ concentrations, the rate of ROS production in HPF rose consistently [45]. Based on these studies and contribution to high oxidative stress in cancer cell lines without immediate cell death effects around 50% (S2 Fig), we chose 2mM $H_2O_2$ as a positive control to observe intracellular ROS levels to be able to compare with other groups (Fig 8). Despite the intracellular ROS observed in positive controls with high $H_2O_2$ and ROS generator concentrations, we observed no protective effect of PtNPs sequestered inside cells against relatively lower $H_2O_2$ and ROS generator concentrations, like

PAW, which agrees with previous results (S2 Fig). Moreover, it suggests that ROS solutions can induce cell death via both membrane damage and intracellular damage caused by ROS influx. This study contributes to a better understanding of potential PAW-induced cell death mechanisms and PtNPs antioxidative behaviour, which will be relevant to improving future applications of PAW and PtNPs for medical use.

## Supporting information

**S1 Fig. Temperature measurement of treated PAW.** In this study, 10 min of plasma treatment was applied to 10 mL of water. The thermal measurement was taken at the end of the treatment.
(TIF)

**S2 Fig. Dose responses for ROS generator and $H_2O_2$.** (A) U-251 MG cells were incubated with 0.032 and 4 μg/ml PtNPs for 4 h and then treated with increasing concentrations ($0 \leq 50$ mM) of ROS generator. (B) U-251 MG cells were incubated with 0.032 and 4 μg/ml PtNPs for 4 h and then treated with increasing concentrations ($0 \leq 8$ mM) of $H_2O_2$. Alamar blue cell viability assay was carried out 4 h after treatment.
(TIF)

**S3 Fig. Reactive species (Nitrate and total oxidative species) measurement of PAW before and after various dilution conditions.** DMEM-DMEM with 10% FBS used for cell culture; Pyr -Pyruvate; ROSg—ROS generator; In this experiment, 30% of PAW was selected as the representative condition. The PAW was diluted in DMEM with or without pyruvate, DMEM without pyruvate but with 4 μg/ml PtNPs and water. 25 mM of ROS generator in DMEM with/without pyruvate and $H_2O$ was also tested as controls.
(TIF)

**S4 Fig. Repeat experiments for dose responses for simultaneous PAW treatment.** U-251 MG cells were treated with 0.032 and 4 μg/ml PtNPs and increasing concentrations ($0 \leq 30\%$) of PAW simultaneously and incubated for 24h **(A)** or 48 h **(B)**.
(TIF)

**S1 Table. ROS and RNS characterisation of PAW generated after 10 min treatment.**
(DOCX)

## Acknowledgments

The authors thank our colleagues in TU Dublin, ESHI and FOCAS institutes who provided technical support and the use of facilities. We also thank Cristina Canal, associate professor of UPC Barcelona, for assistance with providing thermal imaging.

## Author Contributions

**Conceptualization:** Sebnem Gunes, Zhonglei He, Evanthia Tsoukou, Sing Wei Ng, Daniela Boehm, Beatriz Pinheiro Lopes, Paula Bourke, Renee Malone, Patrick J. Cullen, Wenxin Wang, James Curtin.

**Formal analysis:** Sebnem Gunes, Zhonglei He, Sing Wei Ng, Paula Bourke, Renee Malone, Patrick J. Cullen, Wenxin Wang, James Curtin.

**Funding acquisition:** James Curtin.

**Investigation:** Sebnem Gunes, Zhonglei He, Patrick J. Cullen, James Curtin.

**Methodology:** Sebnem Gunes, Zhonglei He, Evanthia Tsoukou, Daniela Boehm, Paula Bourke, Patrick J. Cullen, Wenxin Wang, James Curtin.

**Project administration:** James Curtin.

**Supervision:** Renee Malone, Patrick J. Cullen, James Curtin.

**Writing – original draft:** Sebnem Gunes, Zhonglei He, Patrick J. Cullen, James Curtin.

**Writing – review & editing:** Zhonglei He, Evanthia Tsoukou, Sing Wei Ng, Daniela Boehm, Beatriz Pinheiro Lopes, Paula Bourke, Renee Malone, Patrick J. Cullen, Wenxin Wang, James Curtin.

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
