## [Decision Letter · Decision Letter 0]

29 Jun 2022

PONE-D-22-14482Cell death induced in Glioblastoma cells by Plasma-Activated-Liquids (PAL) is primarily mediated by membrane lipid peroxidation and not ROS influxPLOS ONE

Dear Dr. Curtin,

Thank you for submitting your manuscript to PLOS ONE. After careful consideration, we feel that it has merit but does not fully meet PLOS ONE’s publication criteria as it currently stands. Therefore, we invite you to submit a revised version of the manuscript that addresses the points raised during the review process.

Author must revise manuscript as per both reviewers comments carefully.

We look forward to receiving your revised manuscript.

Kind regards,

Nagendra Kumar Kaushik, PhD

Academic Editor

PLOS ONE

Journal Requirements:

[This work is supported by TU DUBLIN Fiosraigh Research Scholarship programme (S.G.), Irish Research Council Government of Ireland Postdoctoral Fellowship Award GOIPD/2020/788 (Z.H., J.C.), Science Foundation Ireland Grant Numbers 14/IA/2626 (P.B, P.C., J.C.) 15/SIRG/3466 (DB), 16/BBSRC/3391 (PB) and 20/US/3678 (PB).]

 [This work is supported by TU DUBLIN Fiosraigh Research Scholarship programme (S.G.), Irish Research Council Government of Ireland Postdoctoral Fellowship Award GOIPD/2020/788 (Z.H., J.C.), Science Foundation Ireland Grant Numbers 14/IA/2626 (P.B, P.C., J.C.) 15/SIRG/3466 (DB), 16/BBSRC/3391 (PB) and 20/US/3678 (PB).]

Reviewers' comments:

Reviewer's Responses to Questions

**Comments to the Author**

1. Is the manuscript technically sound, and do the data support the conclusions?

Reviewer #1: Yes

Reviewer #2: Partly

2. Has the statistical analysis been performed appropriately and rigorously? 

Reviewer #1: Yes

Reviewer #2: No

3. Have the authors made all data underlying the findings in their manuscript fully available?

Reviewer #1: Yes

Reviewer #2: Yes

4. Is the manuscript presented in an intelligible fashion and written in standard English?

Reviewer #1: Yes

Reviewer #2: Yes

5. Review Comments to the Author

Reviewer #1: In this manuscript, the authors try to identify potential antagonistic effect between antioxidative intracellularly accumulated platinum nanoparticles (PtNPs) and PAL. It is found that PAL can significantly reduce the viability of glioblastoma U-251MG cells and it did not involve measurable ROS influx but instead lead to lipid damage on the plasma membrane of cells exposed to PAL. In addition, it is found that the intracellular antioxidative PtNPs showed no protective effect against PAL. This study contributes to further understanding of principle cell killing routes of PAL and discovery of potential PAL-related therapy and methods to inhibit side effects. This obtained results are very interesting. The paper is well organized and nice written. Thus I would recommend publish it after the authors address several of my minor comments:

1)Since this paper is about using plasma jet for cancer applications, there are several nice review paper on plasma jet and cancer therapy such as (1) Phys. Rep. 630, 1-84 (2016), (2) Int. J. Cancer 134, 1517 (2014), and (3) Scientific Reports, 7, 4479, (2017).(4) Phys. Plasmas 28, 100501 (2021) should be cited and discussed.

2)What's the diameter of the tip of the electrode?

3)What’s temperature of the liquid after plasma treatment?

4)Why choose the deionised water rather than culture medium?

5)What’s the NO2-, NO3-, H2O2 and pH of the PAW?

6)Some of the figures are not clear, please improve it.

Reviewer #2: The paper deals with a study that is aimed to demonstrate if the cell-death mediated by Plasma Activated Water is due to ROS influx or membrane lipid peroxidation. For this purpose, the authors studied the use of PtNPs as potential intracellular ROS scavengers that showed their efficacy in previous experiments performed with direct plasma treatment instead of PAW application. The paper demonstrates that the presence of intra-PtNPs is not efficacious against ROS when cels are exposed to PTW. The paper in the opinion of the referee shows several experiments well described both in term of their rationale and results obtained but there are in the opinion of the reviewer some issues that need to be included in the paper that allow the author to better address their discussion and conclusion. Here below are reported a list of comments, suggestions:

1) The authors used the PAW whose chemical characterization is included in table S1, but diluted in cell culture medium w/o pyruvate. Did the author quantify the concentration of H2O2, NO2- and NO3- after dilution in the cell culture? As well known the cell culture medium could actively contribute to scavange ROS as well. About that, the reviewer suggests to cite “Antioxidants 2021, 10(4), 605; https://doi.org/10.3390/antiox10040605” and similar papers showing the scavenging effect of cell culture media w/o FBS. Thus, in order to be sure that the observed results are connected to a successful scavenging of ROS by antioxidants used or that the use of ROS generator is efficacious in promoting the formation of ROS the reviewer suggests to include quantification of ROS before and after each step (i.e. before and after dilution, before and after addition of ROS generator, before and after addition of pyruvate etc. …)

2) In the opinion of the reviewer before addressing some conclusion about intracellular scavenging of ROS by PtNPs it is necessary to demonstrate the presence of such particles inside cells. Did the authors observe PtNPs inclusion?

3) In the experiments in which the PtNPs are incubated together PAW are the authors sure that the nanoparticles are not affected in their structure by ROS. This aspect could, in the opinion of the referee compromise their efficacy in ROS scavenging. Also in this case it is necessary to quantify the concentration of ROS.

4) Figure 4A and 4B: how the authors explain the data collected at PAW10%? These data in fact have a different trend respect the others because it seems that the addition of PtNPs decreases instead of increasing the cell viability.

5) The statistical analysis of all data collected except for that one shown in figure 2 is not included. The reviewer suggests to insert statistical analysis of all reported data.

6) Can the authors specify why used the concentration of H2O2 reported for the experiments reported in figure 8?

6. PLOS authors have the option to publish the peer review history of their article (what does this mean?). If published, this will include your full peer review and any attached files.

Reviewer #1: No

Reviewer #2: No

---

## [Author Response · Author response to Decision Letter 0]

12 Aug 2022

Dear Editor and Reviewers,

Thank you very much for your comments and suggestions. We have found these to be very helpful to further improve our manuscript. We have addressed every comment in the attached Response to Reviewers document. We have also updated the manuscript accordingly and colour coded the changes in the marked up version of the manuscript. You can find the point by point response (colour coded) in our attached response to reviewers document.

With best wishes,

James, Sebnem and co-authors

---

## [Decision Letter · Decision Letter 1]

30 Aug 2022

Cell death induced in Glioblastoma cells by Plasma-Activated-Liquids (PAL) is primarily mediated by membrane lipid peroxidation and not ROS influx

PONE-D-22-14482R1

Dear Dr. Curtin,

We’re pleased to inform you that your manuscript has been judged scientifically suitable for publication and will be formally accepted for publication once it meets all outstanding technical requirements.

Kind regards,

Nagendra Kumar Kaushik, PhD

Academic Editor

PLOS ONE

Additional Editor Comments (optional):

I have personally checked this revised manuscript and author responses as Reviewer 2 is not responding to verify revision. I recommend accepting this manuscript at this stage as author already revised it carefully and properly.

Reviewers' comments:

Reviewer's Responses to Questions

**Comments to the Author**

1. If the authors have adequately addressed your comments raised in a previous round of review and you feel that this manuscript is now acceptable for publication, you may indicate that here to bypass the “Comments to the Author” section, enter your conflict of interest statement in the “Confidential to Editor” section, and submit your "Accept" recommendation.

Reviewer #1: All comments have been addressed

2. Is the manuscript technically sound, and do the data support the conclusions?

Reviewer #1: Yes

3. Has the statistical analysis been performed appropriately and rigorously? 

Reviewer #1: Yes

4. Have the authors made all data underlying the findings in their manuscript fully available?

Reviewer #1: Yes

5. Is the manuscript presented in an intelligible fashion and written in standard English?

Reviewer #1: Yes

6. Review Comments to the Author

Reviewer #1: The authors addressed all my comments. The obtained results are interesting. I would recommend publish it as is

7. PLOS authors have the option to publish the peer review history of their article (what does this mean?). If published, this will include your full peer review and any attached files.

Reviewer #1: **Yes: **XinPei Lu

---

## [Editor Report · Acceptance letter]

12 Sep 2022

PONE-D-22-14482R1 

Cell death induced in Glioblastoma cells by Plasma-Activated-Liquids (PAL) is primarily mediated by membrane lipid peroxidation and not ROS influx 

Dear Dr. Curtin:

I'm pleased to inform you that your manuscript has been deemed suitable for publication in PLOS ONE. Congratulations! Your manuscript is now with our production department. 

Kind regards, 

on behalf of

Prof. Nagendra Kumar Kaushik 

Academic Editor

PLOS ONE